# Lamin-Related Congenital Muscular Dystrophy Alters Mechanical Signaling and Skeletal Muscle Growth

**DOI:** 10.3390/ijms22010306

**Published:** 2020-12-30

**Authors:** Daniel J. Owens, Julien Messéant, Sophie Moog, Mark Viggars, Arnaud Ferry, Kamel Mamchaoui, Emmanuelle Lacène, Norma Roméro, Astrid Brull, Gisèle Bonne, Gillian Butler-Browne, Catherine Coirault

**Affiliations:** 1Center for Research in Myology, Sorbonne Université, INSERM UMRS_974, 75013 Paris, France; D.J.Owens@ljmu.ac.uk (D.J.O.); j.messeant@institut-myologie.org (J.M.); arnaud.ferry@upmc.fr (A.F.); kamel.mamchaoui@upmc.fr (K.M.); astrid.brullcanagueral@nih.gov (A.B.); g.bonne@institut-myologie.org (G.B.); gillian.butler-browne@upmc.fr (G.B.-B.); 2Research Institute for Sport and Exercise Science, Liverpool John Moores University, Liverpool L3 3AF, UK; m.viggars@2014.ljmu.ac.uk; 3Inovarion, 75005 Paris, France; sophie.moog@inovarion.com; 4Université de Paris, 75006 Paris, France; 5Neuromuscular Morphology Unit, Institute of Myology, Pitié-Salpêtrière Hospital, 75013 Paris, France; e.lacene@institut-myologie.org (E.L.); nb.romero@institut-myologie.org (N.R.); 6APHP, Reference Center for Neuromuscular Disorders, Pitié-Salpêtrière Hospital, Institute of Myology, 75013 Paris, France

**Keywords:** mechanotransduction, muscle growth, nuclear envelope, satellite cell, YAP

## Abstract

Laminopathies are a clinically heterogeneous group of disorders caused by mutations in the *LMNA* gene, which encodes the nuclear envelope proteins lamins A and C. The most frequent diseases associated with *LMNA* mutations are characterized by skeletal and cardiac involvement, and include autosomal dominant Emery–Dreifuss muscular dystrophy (EDMD), limb-girdle muscular dystrophy type 1B, and *LMNA*-related congenital muscular dystrophy (*LMNA*-CMD). Although the exact pathophysiological mechanisms responsible for *LMNA*-CMD are not yet understood, severe contracture and muscle atrophy suggest that mutations may impair skeletal muscle growth. Using human muscle stem cells (MuSCs) carrying *LMNA*-CMD mutations, we observe impaired myogenic fusion with disorganized cadherin/β catenin adhesion complexes. We show that skeletal muscle from *Lmna*-CMD mice is unable to hypertrophy in response to functional overload, due to defective fusion of activated MuSCs, defective protein synthesis and defective remodeling of the neuromuscular junction. Moreover, stretched myotubes and overloaded muscle fibers with *LMNA*-CMD mutations display aberrant mechanical regulation of the yes-associated protein (YAP). We also observe defects in MuSC activation and YAP signaling in muscle biopsies from *LMNA*-CMD patients. These phenotypes are not recapitulated in closely related but less severe EDMD models. In conclusion, combining studies in vitro, in vivo, and patient samples, we find that *LMNA*-CMD mutations interfere with mechanosignaling pathways in skeletal muscle, implicating A-type lamins in the regulation of skeletal muscle growth.

## 1. Introduction

Skeletal muscle is a highly organized tissue designed to produce force and movement. It is largely composed of differentiated, multinucleated, postmitotic myofibers responsible for contraction, and also contains a population of mononucleated muscle stem cells (MuSCs), called satellite cells, that reside between myofibers and the surrounding basal lamina and that display long-term quiescence. Following muscle injury, during postnatal growth and in response to many hypertrophic responses, MuSCs are activated and undergo a highly orchestrated series of events that regulate their proliferation, polarity, and differentiation (reviewed in [1]). Although a subset of MuSCs return to quiescence [2], other activated MuSCs subsequently differentiate and fuse to each other or to existing myofibers. Adhesive contacts between activated MuSCs or between MuSCs and the myofibers are critical to sense and transduce intracellular forces between cells and the extracellular matrix [3,4] and neighboring cells [5,6,7], and provide direct signaling cues essential to stem cell behavior [8]. 

Apart from cell adhesive components, recent studies clearly establish that the nucleus is critical for cells to sense and respond to the mechanical properties of their environment [9,10], thus implicating that muscle plasticity depends on nuclear mechanotransduction. The mechanical properties of the nucleus are largely determined by the nuclear lamina, a fibrous meshwork composed of lamin intermediate filament proteins that underlies the inner nuclear membrane. Nuclear lamins are encoded by three genes; lamin-A and lamin-C (known as A-type lamins) are alternatively spliced products of the *LMNA* gene, whereas lamin-B1 and lamin-B2 (B-type lamins) are encoded by the *LMNB1* and *LMNB2* genes. Mutations in the *LMNA* gene cause laminopathies, a phenotypically diverse group of disorders, including muscular dystrophies and cardiomyopathies [11]. The majority of *LMNA* mutations cause the autosomal dominant Emery–Dreifuss muscular dystrophy or EDMD, characterized by progressive muscle wasting, contractures, and cardiomyopathy. Lamin-related congenital muscular dystrophy (*LMNA*-CMD) manifests as a particularly severe skeletal muscle phenotype, with muscle wasting beginning very early in life [12], frequent nuclear defects [13], and impaired mechanosensing [14].

Cellular mechanotransduction involves physical connection of the nuclear lamina to the cytoskeleton [15]. Such connections are mediated by the members of nucleoskeleton and cytoskeleton (LINC complex) [16] that comprise SUN domain proteins that bind via their nucleoplasmic domains to A-type lamins [16] and nesprins at the outer nuclear membrane that bind to the cytoskeleton [17]. Together A-type lamins and LINC complexes are crucial for mechanical coupling between the nucleoskeleton and the cytoskeleton [10,15]. Functional loss in A-type lamins alters cytoskeletal actin structures around the nucleus in cells cultured on a rigid substrate [18,19,20], presumably through an impaired activation of the mechanosensitive transcriptional cofactor serum responsive factor (SRF) and its target genes [21]. *LMNA*-CMD mutations also compromise the ability of cells to adapt their actin cytoskeleton to different cellular microenvironments and to withstand mechanical stretching of the extracellular matrix, owing to the deregulation of yes-associated protein (YAP) [14], a cotranscriptional factor that nuclear or cytoplasmic localization is modulated by diverse biomechanical signals from the actin cytoskeleton [22]. Collectively, these results implicate A-type lamins in modulating the dynamics and organization of the actin cytoskeleton and thus are also involved in cellular mechanotransduction. 

It is currently unknown whether mechanotransduction defects in *LMNA*-CMD mutations may explain abnormal skeletal muscle growth seen in laminopathic patients. In the current study, we aim to investigate the role of A-type lamins in the regulation of mechanotransduction at cell–cell adhesions and in multinucleated muscle cells. We analyzed three different human cell lines with *LMNA* mutations responsible for congenital muscle dystrophy, namely, *LMNA* c.94_96delAAG, *LMNA* p.Arg249Trp, and *LMNA* p.Leu380Ser. All these mutations are localized in the head (*LMNA* c.94_96delAAG) or in the rod domain (p.Arg249Trp and p.Leu380Ser) of the A-type lamin and are predicted to modify the oligomerization state of the proteins. In other words, these mutations affect the structure and integrity of the nucleoskeleton ([23], compromise the mechanical properties of the nucleus, and force transmission between the nucleoskeleton and the cytoskeleton [15,23]. We also want to determine the consequences of A-type lamin mutations on in vivo muscle adaptation to a mechanical challenge. We hypothesize that *LMNA*-CMD mutations impair cellular and molecular mechanisms contributing to skeletal muscle growth. For the first time, we show that *LMNA*-CMD mutations impaired myogenic fusion in vitro due to disorganized cadherin/β-catenin complexes with reduced M-cadherin and β-catenin protein expression. Defective skeletal muscle growth was also revealed in vivo, since the *Lmna*-CMD mouse model was unable to hypertrophy due to defective accretion of activated satellite cells in response to functional overload. Moreover, myotubes and muscle fibers with *LMNA*-CMD mutations demonstrate aberrant regulation of YAP nucleocytoplasmic translocation in response to different mechanical challenges, which may explain the reduced protein synthesis in mutant fibers. More importantly, in a human context, we reported consistent defects in satellite cell fusion and YAP signaling in muscle sections from *LMNA*-CMD patients, suggesting that defects in mechanosignaling can contribute to the impaired skeletal muscle growth observed in *LMNA*-CMD patients. Overall, our data highlight a critical role of A-type lamins in modulating satellite cell fate through mechanoresponsiveness, and as a consequence, skeletal muscle growth.

## 2. Results

### 2.1. LMNA-CMD Muscle Stem Cells Exit the Cell Cycle but Exhibit Impaired Fusion

We first examined the functional consequences of *LMNA*-CMD mutations on human MuSCs differentiation. To do this, confluent human wild-type (WT) and mutant MuSCs with heterozygous *LMNA* p.Lys32del (*LMNA*^ΔK32^), *LMNA* p.Arg249Trp (*LMNA*^R249W^), and *LMNA* p.Leu380Ser (*LMNA*^L380S^) mutations were shifted from proliferation to differentiation medium (Figure 1A). We observed a severe reduction in the fusion index in all three *LMNA*-CMD mutant cell lines compared with WTs (Figure 1A,B,E).

However, *LMNA*-CMD mutated MuSCs were able to arrest cell division and to express myogenin, an early marker for the entry of MuSCs into the differentiation pathway (Figure 1C,D). No fusion defect was reported in EDMD (*LMNA*^H222P^) mutant cells (Appendix A), i.e., cells with a *LMNA* mutation that manifests as a less severe phenotype. 

### 2.2. Impaired Cell–Cell Interactions in LMNA-CMD Mutant Muscle Cells Precursors

Fusion defects observed in *LMNA*-CMD cells prompted us to examine the pattern of cadherin and catenin-based cell adhesion complexes. We immunostained cadherin and β-catenin in confluent MuSCs (Figure 2A and Figure 3A). In WT MuSCs, cadherin and β-catenin depict the typical “zipperlike” staining pattern at cell–cell contacts, characteristic of force-dependent engagement of the cadherin–catenin complex and cell–cell cohesion (Figure 2A and Figure 3A). In MuSCs expressing ΔK32, R249W and L380S lamin A/C mutations, both cadherin and β-catenin staining was disorganized with a loss of the “zipperlike” staining pattern compared to WT cells (Figure 2A and Figure 3A). In addition, the size of the β-catenin complex was significantly smaller in *LMNA*-CMD mutant cells compared to WTs (Figure 3B, each *p* < 0.001). In agreement with the morphological differences, Western blot quantification showed that mean protein levels of the muscle-specific cadherin, M-cadherin, and β-catenin were also significantly lower in confluent *LMNA*-CMD mutant MuSCs compared to WT (Figure 2B and Figure 3C). Because the cadherin level is primarily regulated through alteration of its stability [24], we further evaluated M-cadherin degradation in the presence of cycloheximide (CHX), an inhibitor of protein synthesis (Figure 2C). In confluent *LMNA*-CMD MuSCs, the presence of CHX in the growth medium resulted in a significant reduction of M-cadherin. In contrast, the protein levels of M-cadherin remained relatively constant in WT MuSCs (Figure 2C). Messenger RNA levels of M-cadherin (*CDH15)* as determined by qRT-PCR were not different between WT and *LMNA*-CMD cells at high cell density (Figure 2D). These findings suggest that reduced levels of M-cadherin in *LMNA*-CMD mutant MuSCs resulted from an increased degradation of M-cadherin at high cell density and not from reduced transcription. Similarly, β-catenin mRNA expression did not differ between WT and *LMNA*-CMD mutant cells (Figure 3D). Collectively, these data indicate that degradation of M-cadherin may contribute to the impaired ability to form cadherin/β-catenin complexes at cell–cell adhesion sites in *LMNA*-CMD mutant MuSCs.

### 2.3. YAP Nuclear Sequestration in LMNA-CMD Mutant Myotubes

In nonmuscle cells, cadherin bound β-catenin at cell–cell contacts is a critical regulator of YAP localization [25,26]. YAP nuclear localization promotes the proliferation of MuSCs whilst inhibiting myogenic differentiation [27]. In static WT myotubes, YAP was predominantly cytoplasmic, as previously reported [27]. In contrast, myotubes with *LMNA*-CMD mutations had a predominant nuclear YAP localization, attesting to a defective localization of YAP (Figure 4A,B). To test whether YAP mislocalization in *LMNA*-CMD was due to reduced M-cadherin protein expression, we treated confluent WT cells with small interfering RNA (siRNA) against M-cadherin and analyzed YAP localization (Figure 4E–G). Depletion of M-cadherin impaired cell-density dependent redistribution of YAP to the cytoplasm, as demonstrated by the significantly higher YAP nucleocytoplasmic ratio in cadherin-depleted WT cells (Figure 4F). These results strongly suggested that altered M-cadherin-/βcatenin/YAP signaling contribute to impaired myogenic fusion in *LMNA*-CMD mutations.

### 2.4. Adaptability to Mechanical Constraints Is Severely Affected in LMNA-CMD Myotubes

As YAP is critical for mechanoresponsiveness, we examined the myotube response to cyclic stretch. We found that whilst WT myotubes were able to respond to mechanical stretch by reorganizing the actin cytoskeleton into parallel actin fibers, the cyclically stretched *LMNA*-CMD mutant myotubes displayed actin fibers that lacked orientation (Figure 4H–J). In addition, cyclic stretching induced relocalization of YAP into the nucleus in WT myotubes whilst nuclear YAP exclusion was observed in stretched *LMNA*-CMD mutant myotubes (Figure 4A–D). This was associated with dephosphorylation of YAP at Ser^127^ in WT but not *LMNA*-CMD mutant myotubes. These data suggest that myotubes carrying *LMNA*-CMD mutations cells are unable to adapt to acute mechanical stretch and to regulate YAP. YAP in turn, regulates and is regulated by the actin cytoskeleton.

### 2.5. Defective Muscle Hypertrophy in Lmna ΔK32 Heterozygous Mice

We next examined the in vivo physiological implications of *LMNA*-CMD-induced muscle defects to sense and respond to mechanical stress. The mouse model carrying the ΔK32 mutation is at present the only characterized *Lmna*-CMD mouse model carrying a human *LMNA* mutation. Following functional overload (FO), we found that the plantaris muscle (PLN) from *Lmna*^+/^^ΔK32^ mice hypertrophied significantly less than their WT counterparts (Figure 5A). Muscle mass normalized to body mass was comparable at baseline (WT = 0.69 ± 0.02 mg·g^−1^ vs *Lmna*^+/^^ΔK32^ = 0.65 ± 0.02 mg·g^−1^). However, normalized PLN mass significantly increased in WT mice after 7 days (1.15 ± 0.07 mg·g^−1^) and 4 weeks (1.59 ± 0.06 mg·g^−1^) of overload, whereas mutant mice showed a significantly impaired adaptive response at both 7 days (0.84 ± 0.05 mg·g^−1^) and 4 weeks (1.27 ± 0.06 mg·g^−1^). In addition, FO significantly increased nuclear deformability in *Lmna*^+/^^ΔK32^ (Appendix A), thus validating in vivo the increased nuclear deformability previously reported in vitro [13]. WT mice also showed significantly improved maximal force production of PLN in response to 4 weeks of overload compared to mutant mice (*p* < 0.05; Figure 5B). In contrast, muscles of the EDMD *Lmna*^H222P^ mice were able to respond to overload to a similar extent to their WT counterparts (Appendix A), suggesting that impaired mechanoresponsiveness and muscle growth are specific to congenital laminopathies.

### 2.6. Defective Myonuclear Accretion in Lmna ΔK32 Heterozygous Mice

Myonuclear accretion is thought to be a determinant of exercise-induced remodeling in skeletal muscle [28] and myonuclear accretion relies on the activation and proliferation of MuSCs, and fusion of the activated MuSCs into new and existing myofibers [29]. MuSC fusion involves the formation of cell–cell contacts, a process regulated by cell–cell adhesion molecules β-catenin and M-cadherin, which we found to be dysregulated in *LMNA*-CMD mutant cells in vitro (Figure 2 and Figure 3). We therefore examined whether MuSCs from *Lmna*^+/^^ΔK32^ mice could be activated and incorporated into new and existing myofibers in response to functional overload (FO). After 1 week of FO, both WT and *Lmna*^+/^^ΔK32^ mutant mice had an increased number of Pax 7^+^ cells (Figure 5C,E) indicating MuSCs proliferation. To determine the fusion capacities of Pax 7^+^ cells, the number of myonuclei (Hoechst staining) inside the sarcolemma (dystrophin immunostaining) were counted in control and mutant PLN muscle sections before and at 1 and 4 weeks after FO. The number of myonuclei per myofiber was similar before FO between WT and mutant muscles and increased significantly in WT PLN muscles following 4 weeks of FO (Figure 5D). Conversely, myonuclei number did not change in *Lmna*^+/^^ΔK32^ mice at 1 and 4 weeks of FO and was significantly lower than in WT after 4 weeks of FO (Figure 5D). Taken together, our data show that *Lmna*^+/^^ΔK32^ mutation leads to a lack of myonuclear accretion in response to functional overload, despite activation and proliferation of MuSCs. 

### 2.7. YAP Abundance Is Higher at Baseline but Decreases after FO in Lmna ΔK32 Heterozygous Mice

To assess whether defective hypertrophy could be a consequence of aberrant mechanoresponsiveness in mature fibers and muscle precursor cells, we analyzed YAP signaling in PLN from *Lmna*^+/^^ΔK32^ mice. It is well established that YAP stimulates muscle fiber hypertrophy and protein synthesis [30,31]. In control conditions, YAP labeling was clearly detectable at the muscle fiber membrane and was also detected in some nuclei localized within the laminin boundary of muscle fibers (Figure 6A), consistent with previous data [30,31]. However, before FO, the number of YAP^+^ fibers was significantly higher in *Lmna*^+/^^ΔK32^ compared to WT mice (*p* < 0.001, Figure 6A,B. After 1 week FO, the number of YAP^+^ fibers significantly increased in the WT mice whilst YAP staining was markedly reduced in *Lmna*^+/^^ΔK32^ mice (Figure 6A,B). Four weeks after the FO procedure, the number of YAP^+^ fibers had returned to baseline in the WT (Figure 6A,B). In the *Lmna*^+/^^ΔK32^ mice, YAP was still downregulated after 4 weeks FO compared to corresponding baseline values. 

### 2.8. Defective Muscle Protein Synthesis in Lmna ΔK32 Heterozygous Mice

We reasoned that the increase in YAP signaling seen in the early stages of remodeling in overloaded WT muscle would be associated with increased rates of protein synthesis. By employing the SUnSET method, which measures acute incorporation of puromycin into newly synthesized peptides [32], we determined the rate of protein synthesis in PLN muscle of CON and FO mice after 7 days of FO. As hypothesized, PL of WT mice responded to FO by increasing rates of protein synthesis (*p* < 0.0001), whereas *Lmna*^+/^^ΔK32^ mice did not (*p* = 0.0344; Figure 6C).

### 2.9. Neuromuscular Junction Defects after Functional Overload in Lmna ΔK32 Heterozygous Mice

Since the neuromuscular junction (NMJ) is an adaptable/plastic synapse, highly sensitive to decreased or increased activity, we next decided to analyze morphological changes occurring at NMJs of WT and *Lmna*^+/^^ΔK32^ mice following 4 weeks of FO. Muscle fibers were isolated from plantaris muscle and stained with α-bungarotoxin (α-BTX) to label acetylcholine receptor (AChR) clusters as well as antibodies against neurofilament (NF) and synaptophysin (Syn) to visualize axon and nerve terminals, respectively (Figure 7). In control condition, both WT and *Lmna*^+/^^ΔK32^ mice exhibited mature NMJs characterized by an elaborate continuous topology that have a pretzel-like shape (Figure 7A). Interestingly, we observed a significant increase in AChR clusters area both in WT (*p* < 0.0005) and *Lmna*^+/^^ΔK32^ (*p* < 0.05) mice after FO, as determined by fluorescent labeling with α-BTX (Figure 7B [i]). The nerve terminal area of both WT and *Lmna*^+/^^ΔK32^ was also significantly increased following FO and pre-postsynaptic overlap was unaffected by FO in both strains of mice (Figure 7B [ii,iii]). However, despite *Lmna*^+/^^ΔK32^ demonstrating an increase in AChR cluster area in response to FO, postsynaptic architecture appeared discontinuous with isolated AChR clusters (Figure 7A). Morphometric analysis revealed that the number of AChR clusters per NMJ (i.e., the number of continuous AChR-stained structures per synapse) was significantly increased in *Lmna*^+/^^ΔK32^ following FO, indicating a severe dismantlement of the postsynaptic counterpart (Figure 7B [iv]). These results suggest NMJ stability is impaired in *Lmna*^+/^^ΔK32^ mice following FO.

### 2.10. Muscle Biopsies from LMNA-CMD Patients Revealed Increased Pax7^+^ Cells and YAP Signaling

Finally, to corroborate our findings from the in vitro patient cells and in vivo mouse models of striated muscle *Lmna*-CMD in a clinically relevant context, we examined skeletal muscle biopsy samples from patients and control donors. Satellite cells expressing the Pax7^+^ cells are known to decline with age, therefore we examined the number of Pax7^+^ cells in an age-dependent context. The number of Pax7^+^ cells was three- to four-fold higher in the two children with *LMNA*-CMD compared to the age-matched control (*p* < 0.001), and nearly 1.5-fold higher to that observed in a control newborn (4 day-old) (Figure 8A–C and Appendix A). More importantly, almost half of Pax7^+^ cells from human *LMNA*-CMD did not reside in the satellite cell position but instead were found in the interstitial space (Figure 8C). These data suggest an increased proliferation of MuSCs in *LMNA*-CMD patient biopsies with a decreased ability to fuse or to return to quiescence. In addition, muscle cryosections were immunofluorescently labeled for YAP. In the control muscle, YAP labeling was clearly detectable in 42 ± 3% fibers and 13 ± 1% nuclei localized within the laminin boundary of muscle fibers (Figure 8D–F). Interestingly, the percentage of YAP^+^ fibers and YAP^+^ nuclei were respectively two- and three-fold higher in the *LMNA*-CMD patients compared with their relative age-matched control (each *p* < 0.01), and slightly lower to what was observed in a control newborn (Appendix A). Taken together, these data highlight a novel mechanism by which defective accretion of activated MuSCs and altered YAP signaling in MuSCs and multinucleated muscle cells contribute to muscle growth defects in the most severe form of the human striated muscle laminopathies.

## 3. Discussion

A-type lamins are major nuclear proteins involved in mechanosensing and signaling between the nucleus, the cytoskeleton, and the extracellular matrix. There is increasing evidence that lamins and nucleocytoskeletal coupling are required for cellular and nuclear mechanotransduction, muscle development, and plasticity. The mechanisms by which *LMNA* mutations result in muscle-specific defects remain unclear, thus preventing the development of effective therapeutic approaches. In a recent study, nuclear defects including nuclear envelope rupture, DNA damage, and chromatin protrusions have been correlated with the severity of muscle laminopathies [13]. It remains unclear how nuclear defects impair muscle differentiation and growth. Here, we describe a process by which *LMNA* mutations responsible for congenital muscle dystrophy impaired the ability of skeletal muscle to hypertrophy in response to a mechanical challenge, thus altering muscle growth. This relies on aberrant mechanoresponsiveness and fusion defects. Thus, our data highlight a critical role of A-type lamins in modulating skeletal muscle growth. 

Mechanical stimuli are transferred to the actin cytoskeleton, leading to the activation of various signaling pathways that alter cellular dynamics and ultimately control key cell fate decisions such as proliferation and growth arrest. The transcriptional coactivator YAP is tightly regulated by the actin cytoskeleton and has been implicated as a main signaling protein in skeletal muscle mechanotransduction [33]. In activated satellite cells and proliferating MuSCs, YAP is predominantly nuclear permitting cell proliferation [27,34]. In a differentiated postmitotic multicellular context, it is the physical and architectural properties of the cellular microenvironment that inform the cell of its capability to growth, a process that is controlled by YAP/TAZ signaling [35]. Indeed, knockdown of YAP results in impaired proliferation presumably by desensitizing the cell to its physical constraints [36]. As MuSCs exit the cell cycle and fuse to form multinucleated myotubes, YAP is phosphorylated by LATS 1/2 kinase and sequestered in the cytoplasm by 14-3-3 proteins, rendering YAP inactive [37]. However, YAP can be reactivated in myotubes by mechanically stretching cells [35] and in adult myofibers by functional overload of muscle [30]. 

To date, all mutations in A-type lamins [38,39] or nesprins [40,41,42] that cause striated muscle disease compromise the nesprin/SUN/lamin interactions, resulting in dysfunctional nucleocytoskeletal linkages [10,39,40,41,43,44]. There is growing evidence that A-type lamins and/or functional nucleocytoskeletal mechanical coupling are required for normal YAP signaling. We have previously shown that human-derived A-type lamin mutant MuSCs and deficient nesprin-1 MuSCs are unable to traffic YAP to the nucleus in response to cyclic strain [14]. Others have demonstrated by traction force mapping that force transfer from the cytoskeleton to the nucleus is dependent on the LINC complex proteins and is critical for YAP trafficking and transcriptional activity [45]. Whether A-type lamins modulate YAP signaling in multinucleated muscle fibers was not yet known. Here, we show that mechanically stretching human primary myotubes results in YAP translocation to the nucleus, whereas laminopathic (ΔK32) mutant myotubes showed an aberrant, mirrored response. In *LMNA* ΔK32 myotubes, YAP was nucleoplasmic in static conditions and was exported from the nucleus following cyclic stretch with a concomitant decrease in the N/C ratio (Figure 4). This YAP signaling defect was accompanied by a lack of actin cytoskeleton remodeling that was observed in WT myotubes following stretch (Figure 4). In addition, we previously reported that stretch-induced severe cytoskeletal damage of *LMNA*-CMD myoblasts promoted nuclear exclusion of YAP [14]. Moreover, WT mice showed a similar response in vivo, with increased YAP^+^ fibers following functional overload. In contrast *Lmna*^+/ΔK32^ control mice had more YAP^+^ fibers at baseline, but the number of YAP+ fibers decreased following functional overload. Importantly, YAP defects appear to be specific to congenital laminopathies, as the Lmna^H222P/H222P^ mouse model of EDMD was able to respond to FO comparably to WT mice (Appendix A). 

A key finding from our study is that muscle progenitor fusion defects are present in *LMNA*-CMD patient-derived muscle cells and in a mouse model carrying an A-type lamin mutation responsible for *LMNA*-CMD. Human muscle biopsies from patients with *LMNA*-CMD have a greater number of Pax7^+^ cells, providing further support for defective incorporation of activated MuSCs into the myofibers. By first implementing in vitro experimentation, we demonstrate that fusion defects exist in *LMNA*-CMD cells and may be due to aberrant adherens junction formation. Adherens junctions are large macromolecular complexes that accumulate at cell–cell contacts, the formation of which requires cadherins and catenins [46,47,48,49]. In our study, M-cadherin and β-catenin were poorly organized in confluent *LMNA*-CMD MuSCs, whereas confluent WT counterparts displayed a typical zipperlike formation, characteristic of force-bearing adherens junctions [50]. Importantly, adherens junctions are crucial for cellular mechanosensitivity, permitting mechanical forces to mediate cellular behavior. In skeletal muscle, cadherin-mediated adhesions contribute to the quiescence of MuSCs in the niche cells by providing structural integrity, mechanosensation, cell polarity, and juxtacrine signaling [51,52]. Disruption of cadherin-based adhesion between MuSCs and the myofiber is a critical step allowing the transition from quiescence to activation and proliferation [52]. One likely mechanism of mechanical activation of MuSCs is that the stretch produced on cadherins (i.e., on cell-cell adhesions) damages their structure and/or their function thus leading to the departure of MuSCs from the niche, and activation [51,52]. Reduced M-cadherin expression in the current study may also contribute to the impaired differentiation and plasticity of A-type lamin mutant muscle. Numerous in vitro studies have shown that M-cadherin is essential for the differentiation of myoblasts into myotubes [46,53,54,55]. Although M-cadherin is necessary for myoblast fusion, it does not appear to affect the induction of myogenesis [56]. Altered cadherin-based adhesion supports our findings that A-type lamin mutant myoblasts were able to induce myogenin protein expression, but displayed a lower fusion index than wild type MuSCs (Figure 1) and may explain the higher number of Pax7^+^ cells present in *LMNA*-CMD patients (Figure 8). Overall, the data presented here support a YAP-related coupling between the nuclear lamina and the M-cadherin-β-catenin complexes that subsequently modulate muscle growth and plasticity. 

Adaptation of skeletal muscle to physical challenges is accompanied by NMJ remodeling, such that both endurance and resistance exercise cause phenotypic changes in pre- and postsynaptic structures [57]. Accordingly, in our study WT and to a limited extent *Lmna*^+/ΔK32^ NMJs increased in size without any change in pre-postsynaptic overlap following FO (Figure 7). However, whereas WT NMJs in overloaded muscle displayed a typical continuous pretzel-like structure, the postsynaptic network of overloaded *Lmna*
^+/ΔK32^ mice was highly fragmented, suggesting a compromised maintenance of NMJs. The molecular mechanisms by which NMJ plasticity is compromised in lamin-mutated muscle may be attributed to YAP deregulation. YAP is a crucial regulator of neuromuscular junction formation and regeneration. In muscle-specific YAP mutant mice, postsynaptic and presynaptic differentiation and function was impaired and subsequently inhibited NMJ regeneration after nerve injury [58]. 

In conclusion, we show here that functional A-type lamins are critical to allow fine tuning of the appropriate mechanosignaling required for skeletal muscle growth. *LMNA* mutations responsible for congenital muscle dystrophy impaired the ability of skeletal muscle to hypertrophy in response to a mechanical challenge due to impaired fusion of satellite cells, aberrant YAP signaling, and impaired neuromuscular junction. Thus, our data highlight a critical role of A-type lamins in modulating skeletal muscle growth. 

## 4. Materials and Methods

### 4.1. Cell Cultures

#### 4.1.1. Human Myoblasts and Cell Culture

Muscle biopsies were obtained from the Bank of Tissues for Research (Myobank, a partner in the EU network EuroBioBank, Paris, France) in accordance with European recommendations and French legislation. Written informed consent was obtained from all patients. Experimental protocols were approved by our institution (INSERM, Paris, France). Experiments were performed using immortalized human myoblasts carrying the following heterozygous *LMNA* mutations responsible for *LMNA*-CMD (hereafter referred to as *LMNA*-CMD): a *LMNA* c.94_96delAAG, p.Lys32del (hereafter referred to as ΔK32), *LMNA* p.Arg249Trp (hereafter referred to as R249W), and *LMNA* p.Leu380Ser (hereafter referred to as L380S) mutation [12]. Immortalization and characterization of the *LMNA*-CMD myoblasts have been previously described [12,59].

Control immortalized myoblasts were obtained from two healthy control subjects without muscular disorders (hereafter referred to WT1 and WT2). Following muscular biopsy, muscle cell precursors were immortalized and cultured in growth medium consisting of 1 vol 199 Medium/4 vol DMEM (Life Technologies, Carlsbad, CA, USA) supplemented with 20% fetal calf serum (Life technologies, Carlsbad, CA, USA), 5 ng/mL hEGF (Life Technologies, Carlsbad, CA, USA), 0.5 ng/mL βFGF, 0.1 mg/mL dexamethasone (Sigma-Aldrich, St. Louis, MO, USA), 50 µg/mL fetuin (Life Technologies, Carlsbad, CA, USA), 5 µg/mL insulin (Life Technologies, Carlsbad, CA, USA), and 50 mg/mL Gentamycin (Gibco™, Life Technologies, Carlsbad, CA, USA). Differentiation was induced by switching confluent myoblasts to differentiation medium containing DMEM (Gibco) and 50 mg/mL gentamycin. 

#### 4.1.2. Immortalized MyoD-Converted Human Myoblasts

EDMD (*LMNA*^H222P^, carrying the heterozygous *LMNA p.H222P* mutation previously described in patient with classical form of EDMD, [60]) and control patient fibroblasts were obtained from skin biopsies and immortalized as previously described [61]. Inducible myogenic conversion was obtained using a doxycycline-inducible Myod1 lentivirus [62]. MyoD-transfected fibroblasts were cultured in a proliferation medium consisting of DMEM, supplemented with 10% fetal bovine serum (Life Technologies) and 0.1% gentamycin (Invitrogen). For myoconversion, doxycycline (2 µg/mL) (Sigma Aldrich) was added in the differentiation medium, composed of DMEM with 10 µg/mL Insulin.

#### 4.1.3. Drug Treatments and siRNA

Eukaryotic translation inhibitor cycloheximide (CHX) (Sigma-Aldrich, St. Louis, MO, USA) was diluted to a final concentration of 30 µg/mL in the culture medium and added to adherent myoblasts for 4 h. The siRNA transfections were done with HiPerfect (Qiagen, Venlo, Netherlands) according to manufacturer’s instructions. Downregulation of M-cadherin was observed 72 h after transfection. Sequences of siRNAs are provided in Table 1.

#### 4.1.4. Cyclic Strain

Cells were plated on Bioflex culture plates (Flexcell International, Burlington, NC, USA) coated with fibronectin, and then stretched (10% elongation, 0.5 Hz, 4 h). Following 4 h stretch, cells were fixed for immunocytochemistry or collected, as described below.

#### 4.1.5. Immunocytochemistry and Image Analysis

Myotubes were fixed for 5 min with 4% formaldehyde, permeabilized with 0.1% Triton X100 and blocked with 10% bovine serum albumin (BSA) diluted in phosphate buffer solution (PBS). Cells were stained with Phalloidin-Alexa 568 to label F-actin (Interchim, Montluçon, France). The following primary antibodies were used for immunostaining: anti-M-cadherin (Abcam, ab65157), anti-pan-cadherin (Abcam, ab6529), anti-YAP/TAZ (Santa-Cruz, sc-10119s), anti-β-catenin (Cell Signaling, cs-9581), and antimyosin (MF20, DSHB). Secondary antibodies (Life Technologies, Saint-Aubin, France; 1/500) were: Alexa Fluor 488 donkey antimouse IgG or Alexa Fluor 488 donkey antimouse IgG. Primary antibodies were incubated overnight. Nuclei were stained with Hoechst (ThermoFischer) and Mowiol was used as the mounting medium. Confocal images were taken with an Olympus FV 1200 (Olympus, Hamilton, Bermuda) or a laser-scanning microscopy Nikon Ti2 coupled to a Yokogawa CSU-W1 head. 

All image analyses were performed using Fiji software (version 1.51, https://fiji.sc/). Quantification of β-catenin areas at cell–cell contacts was determined in at least 5 different fields for each experimental condition. For YAP analysis, Z-stacks of images were acquired for each channel, and the middle confocal slice was chosen from the images of the nucleus detected in the Hoechst channel. On the corresponding slice in the YAP channel, the average fluorescence intensity in the nucleus and just outside the nucleus (cytoplasm) was measured to determine the nuclear/cytoplasmic ratio. Fusion index was defined as the number of myosin heavy chain expressing myotubes with greater than 2 nuclei divided by the total number of nuclei. 

#### 4.1.6. SDS-PAGE and Protein Analysis

Cells were lysed in total protein extraction buffer (50 mM Tris-HCl, pH 7.5, 2% SDS, 250 mM sucrose, 75 mM urea, 1 mM DTT) with added protease inhibitors (25 μg/mL Aprotinin, 10 μg/mL Leupeptin, 1 mM 4-[2-aminoethyl]-benzene sulfonylfluoride hydrochloride, and 2 mM Na_3_VO_4_) or directly in 2× Laemmli buffer. Twenty milligrams of total protein were loaded into a 10% or a 12% SDS polyacrylamide gel (BioRad, Hertfordshire, UK). Electrophoresis was performed for 30 min at 250 V. Proteins were then transferred onto PVDF or nitrocellulose membranes. After blocking for 1 h with bovine serum albumin (BSA, 5%), membranes were incubated overnight with the following primary antibodies: anti-YAP (1/1000, Santa-Cruz, sc-10119), anti-M-cadherin (1/1000, Abcam, ab-65157), anti-β-catenin (1/1000, cs-9581), or anti-GAPDH (1/10000, Cell Signaling, cs-2118). Secondary antibodies raised again goat antimouse, goat antirat or donkey antigoat HRP conjugates were incubated for 1 h at room temperature with gentle agitation. Detection of adsorbed HRP-coupled secondary antibodies was performed by ECL reaction with Immobilon Western chemiluminescent HRP substrate (Millipore, Billerica, MA, USA). Membranes were washed 3 times for 10 min with tris-buffered saline-tween (TBS-T). HRP signals were detected using a CCD-based detection system (Vilber Lourmat, Marne-la-Vallée, France) or a G-box system with GeneSnap software (Ozyme, Saint-Quentin, France). Membranes subjected to a second round of immunoblotting were stripped with stripping buffer (62.5 mM Tris-HCL pH 6.8, 2% SDS, 100 mM β-mercaptoethanol) and incubated at 55 °C for 30 min with mild shaking before excessive washing with deionized water and reblocking. Quantification was performed using ImageJ (1.X, https://imagej.net/Welcome). 

#### 4.1.7. Quantification of Gene Expression

The mRNA was isolated from cell lysates using the RNeasy mini kit (Qiagen, Hilden, Germany) with the Proteinase K step, according to the manufacturer instruction. The complementary DNA (cDNA) was transcribed by SuperscriptIII (Life Technologies, Carlsbad, CA, USA). Gene expression was quantified by using PerfeCTa-SYBR^®^ Green SuperMix (Quanta Biosciences, Gaithersburg, MD, USA) with the help of LightCycler 480 II (Roche Diagnostics GmbH, Mannheim, Germany). The primers were designed by Primer-BLAST (NCBI) and synthesized by Eurogentec (Liège, Belgium). Expression of all target genes was normalized to the expression of the reference gene *RPLP0*. Primer sequences are listed in Table 1.

### 4.2. Animal Study

#### 4.2.1. Animals

All animal experiments were conducted in accordance with the European Guidelines for the Care and Use of Laboratory Animals and were approved by the institutional ethics committee (APAFIS#2627-2015110616046978, 9 May 2016). All experiments were performed on male mice. Accredited personnel dedicated to the Care and Use of Experimental Animals conducted all animal experiments (accreditation numbers #75–1102 and #75–786). *Lmna*^+/ΔK32^ (*n* = 14) and WT C57Bl/6_129/6J (*n* = 21) littermates were 3 months old at the beginning of the experiments. *Lmna*^+/ΔK32^ mice in a C57Bl/6_129/SvJ genetic background were generated by homologous recombination as described previously [63]. The heterozygous *Lmna*^+/ΔK32^ mouse was chosen over homozygous mice as it is the same mutation seen in *LMNA*-CMD patient, increasing the translational potential of the data derived from this model. Additional experiments were performed in the 129S2/SvPasCrl *Lmna*^H222P/H222P^ mice (*n* = 14), the mouse model of the classical form of EDMD [64] and compared them with their respective control strains (WT 129S2/SvPasCrl, *n* = 12).

#### 4.2.2. Functional Overload

*Lmna*^+/ΔK32^, *Lmna*^H222P/H222P^, and their respective control strain mice were used in the study and assigned to overload (FO) or control (CON) (*n* = 11 WT C57Bl/6_129/6J; *n =* 6 WT 129S2/SvPasCrl; *n* = 7 *Lmna*^+/ΔK32^) groups. Functional overload (FO) of plantaris (PLN) muscles of WT (*n* = 10 WT C57Bl/6_129/6J and *n* = 6 WT 129S2/SvPasCrl) and mutant (*n* = 7 *Lmna*^+/ΔK32^ and *n* = 7 in *Lmna*^H222P/H222P^) mice was induced through the tenotomy of soleus and gastrocnemius muscles, in both legs [65]. The muscles were then sutured from the distal tendon to the proximal musculotendinous region leaving the plantaris intact. Animals recovered within 1–2 h following the end of the procedure and were then monitored daily following surgery for signs of discomfort and infection. For pain management, buprenorphine was administered prior to and following surgery (Vetergesic^©^ 0.3 mg/mL, SC: 0.10 mg/kg). At the indicated time (1 and 4 weeks after FO), animals were sacrificed by cervical dislocation and PLN muscles were dissected and processed for molecular or histological analyses. Following removal of visible fat and connective tissue, isolated PLN muscles were quickly frozen in liquid nitrogen cooled isopentane for cryosectioning or snap frozen in liquid nitrogen or fixed in 4% PFA at room temperature for one hour for analysis of the neuromuscular junction and single muscle fibers. 

#### 4.2.3. In Vivo Estimation of Protein Synthesis

Protein synthesis was measured using the SUnSET method as previously described [32]. In brief, the mice were injected with puromycin (reconstituted in PBS) at a dose of 0.04 µmol·g^−1^ body weight via an intraperitoneal injection exactly 30 min before experimental end point. Muscles were lysed via mechanical disruption in Roche MagnaLyser tubes containing ceramic beads (Roche, Germany) and ice cold RIPA buffer. Total cell lysate protein content was determined via a BCA protein assay (Pierce, Lancashire, UK). Twenty milligrams of total protein were loaded into a 12% stain-free polyacrylamide gel (BioRad, Hertfordshire, UK). Electrophoresis was performed for 30 min at 250 V. Proteins were then transferred onto nitrocellulose membranes. Membranes were blocked for 1 h with BSA (5%) then probed for puromycin with a mouse monoclonal puromycin antibody, clone 12D10 (1:20,000 in 5% BSA, Merck Millipore, Guyancourt, France) for 1 h at room temperature with gentle agitation. The following day, membranes were washed 3 times for 10 min with TBS-T. A secondary horseradish peroxidase antibody raised against the same species as the primary antibody was then applied to membranes (1:2000 in 5% BSA, Merck Millipore) for 1 h at room temperature with gentle agitation. Membranes were washed 3 times for 10 min with TBS-T and then exposed to a chemiluminescent substrate and imaged on a Bio-Rad Chemi-Doc MP. Total puromycin was calculated relative to total protein.

#### 4.2.4. Maximal Force Measures

Maximal isometric tension of the PLN muscle was assessed in situ in response to nerve stimulation, as described previously [66]. Briefly, the knee and foot were secured with pins and the distal tendon of the PLN was attached to a lever arm of a servomotor (305B Dual-Mode Lever, Aurora Scientfic, Aurora, ON, Canada) with silk ligature. 

#### 4.2.5. Immunohistochemistry

Transverse serial sections (8–10 µm) of PLN muscles were obtained using a cryostat, in the mid-belly region. For determination of muscle fiber cross sectional area and minimal Feret diameter, sections were stained with an antidystrophin antibody (MANDYS8(8H11) Developmental Studies Hybridoma Bank, University of Iowa, USA) to label the myofiber border. Additional sections were stained for laminin (Dako, Z0097), YAP (Santa-Cruz, sc-10119), and/or Pax7 (Developmental Studies Hybridoma Bank, University of Iowa, USA). Multiple images were captured of each section using the tile scanning feature on a Leica DM6000 fluorescence wide-field transmission microscope, allowing imaging of the entire section. Myonuclear counts were achieved using an unbiased automated approach; tile-scanned sections stained for dystrophin and DAPI were coded by one member of the research team and analyzed by Myovision software version 5.0 (www.MyoVision.org) [67] by another member of the team. Myonuclei are defined by the software as any nuclear region having its centroid and greater than 50% of its area inside the sarcolemma.

For the analysis of YAP-positive nuclei and YAP-positive fibers, frozen unfixed sections were blocked 1 h in PBS plus 2% BSA and 2% sheep serum. Sections were then incubated overnight with primary antibody against YAP (Santa Cruz, Dallas, TX, USA). After washes in PBS, sections were incubated for 1 h with secondary antibodies (Alexa fluor, Life Technologies, Saint-Aubin, France). YAP-positive nuclei and YAP-positive fibers were counted from thresholded images using Fuji. A minimum of 2000 individual fibers were analyzed from 5 different mice per experimental condition.

Neuromuscular junction (NMJ) analysis was performed on isolated muscle fibers as previously described with minor modifications [68]. Briefly, plantaris muscles were dissected and fixed in 4% PFA/PBS for 1 h and rinsed with PBS at room temperature. Isolated muscle fibers were washed three times for 15 min in PBS, incubated for 30 min with 100 mM glycine in PBS, and rinsed in PBS. Samples were permeabilized and blocked in blocking buffer (4% BSA/5% goat serum/0.5% Triton X-100 in PBS) for 4 h at room temperature. They were then incubated overnight at 4 °C with rabbit polyclonal antibodies against 68 kDa neurofilament (NF, Millipore Bioscience Research Reagents, Guyancourt, France, 1:1000) and synaptophysin (Syn, Thermofisher Scientific, 1:750) in blocking buffer. After four 1 h washes in PBS, muscles were incubated overnight at 4 °C with Cy3-conjugated goat antirabbit IgG (Jackson Immunoresearch Laboratories, Cambridgeshire, UK, 1:500) and Alexa Fluor 488-conjugated α-bungarotoxin (α-BTX, Life Technologies, 1:500) in blocking buffer. After four 1 h washes in PBS, isolated muscle fibers were then flat-mounted in Vectashield (Vector Laboratories, Peterborough, UK) mounting medium. Confocal images were acquired using Leica SPE confocal microscope with a Plan Apo 63x NA 1.4 oil objective (HCX, Leica). Confocal software (LAS AF, Leica, Nußloch, Germany) was used for acquisition of Z serial images, with a Plan Apo 63× NA 1.4 oil objective (HCX, Leica). Confocal images presented are single-projected images derived from image stacks. For all imaging, exposure settings were identical between compared samples and groups. Quantifications were done using ImageJ software. AChR cluster area corresponds to the occupied area of α-BTX fluorescent labeling. More than 20 fibers from at least five different mice of each group were analyzed.

### 4.3. Human Study

Human muscle sections were obtained from 2 *LMNA*-CMD patients, 1 EDMD patient, and 3 control subjects without any muscular disorder. All patients provided informed consent. Clinical summaries and muscle characteristic of all patients are provided in Table 2. All the patients underwent an open muscle biopsy for morphological, immunochemical, and biochemical analyses on snap-frozen muscle tissue. Transverse serial sections (8–10 µm) were stained with for laminin (Dako, Z0097), YAP (Santa-Cruz, sc-10119), and/or Pax7 (Developmental Studies Hybridoma Bank, University of Iowa, USA). Multiple images were captured of each section using the tile scanning feature on a Leica DM6000 fluorescence wide-field transmission microscope, allowing imaging of the entire section.

At the time of muscle biopsy, all patients exhibited common features that are characteristics of *LMNA*-CMD patients, i.e., severe muscle wasting and weakness, no or poor head and trunk control, and respiratory failure. C1 to C3: controls; muscle biopsies in control subjects did not reveal any defect.

### 4.4. Statistical Analysis

Graphpad Prism (version 9, Graphpad Software, La Jolla, CA, USA) was used to calculate and plot mean and standard error of the mean (SEM). All statistical significances were assessed by ANOVA followed by Bonferroni for multiple tests or two-tailed unpaired t-tests. Differences between conditions were considered significant at *p* < 0.05. In case of no difference between WT1 and WT2, pooled values of WT (WT1 and WT2) were presented. Figures were plotted using Graphpad Prism and R with ggplot2 [69].

## Figures and Tables

**Figure 1 ijms-22-00306-f001:**
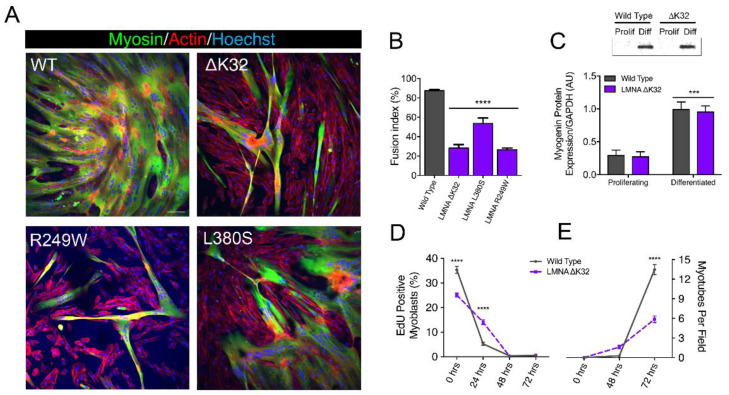
In vitro myoblast fusion and myotube formation. (**A**) Confocal immunofluorescence images of myosin (MF20, green) in wild-type (WT) and *LMNA*-related congenital muscular dystrophy *(LMNA*-CMD) mutant (ΔK32, L380S and R249W) cells, after 3 days of differentiation. Nuclei are stained with Hoechst (blue). Scale bar = 100 µm. (**B**). Fusion index in WT and *LMNA*-CMD mutant cells after 3 days of differentiation. Pooled values of WT (WT1 and WT2) are presented. *n* = 10 fields of view analyzed per time point from across 3 separate experiments. Values are expressed as means ± SEM. **** *p* < 0.0001 versus WT myotubes. (**C**) Myogenin expression in WT and *LMNA* ΔK32 mutant cells in proliferation and after 3 days of differentiation. *n* ≥ 3 from at least 2 separate experiments. *** *p* < 0.001 versus WT myotubes. (**D**) EdU positive myoblasts (%) and (**E**) number of myotubes per field until 3 days of differentiation. *n* = 10 fields of view analyzed per time point from across 3 separate experiments. Values are expressed as means ± SEM. **** *p* < 0.0001 versus WT cells.

**Figure 2 ijms-22-00306-f002:**
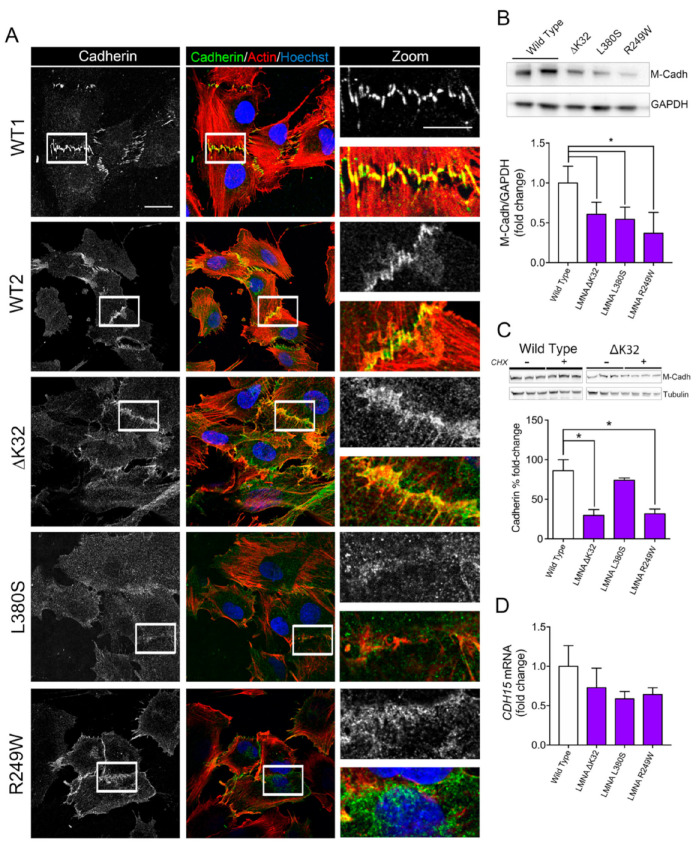
Cadherins in wild-type and mutant muscle cell precursors. (**A**). Confocal immunofluorescence images of F-actin (phalloidin, red) and cadherin (white or green) in WT (WT1 and WT2) and *LMNA*-CMD mutant (ΔK32, L380S and R249W) muscle cell precursors. Nuclei are stained with Hoechst (blue). Scale bar: 20 µm. Zoomed region of cell-cell junctions are shown in left panels. Scale bar: 10 µm. (**B**). Top: Representative Western blot of M-cadherin and GAPDH expression in WT and *LMNA* mutant myoblasts. Bottom: Quantification of M-cadherin protein levels normalized to GAPDH and expressed as fold change versus WT. Values are means ± SEM, *n* ≥ 3 from at least 2 separate experiments. * *p* < 0.005 compared with WT. (**C**) Top: Representative Western blot M-cadherin and α-tubulin expression in WT and R249W myoblasts after 4h-treatment with cyclohexamide (CHX). Bottom: Percentage fold-change in M-cadherin protein levels in WT and mutant myoblasts after CHX treatment. M-cadherin protein levels normalized to β-tubulin. Pooled values of WT (WT1 and WT2) are presented. Values are means ± SEM, *n* = 3 in WT and mutant cell lines. * *p* < 0.05 compared with WT. (**D**) mRNA expression of *CDH15* normalized to RPLP0 and expressed as fold-changes vs WT. Pooled values of WT (WT1 and WT2) are presented. Values are means ± SEM, *n* = 3 separate experiments. There was no significant difference between cell lines.

**Figure 3 ijms-22-00306-f003:**
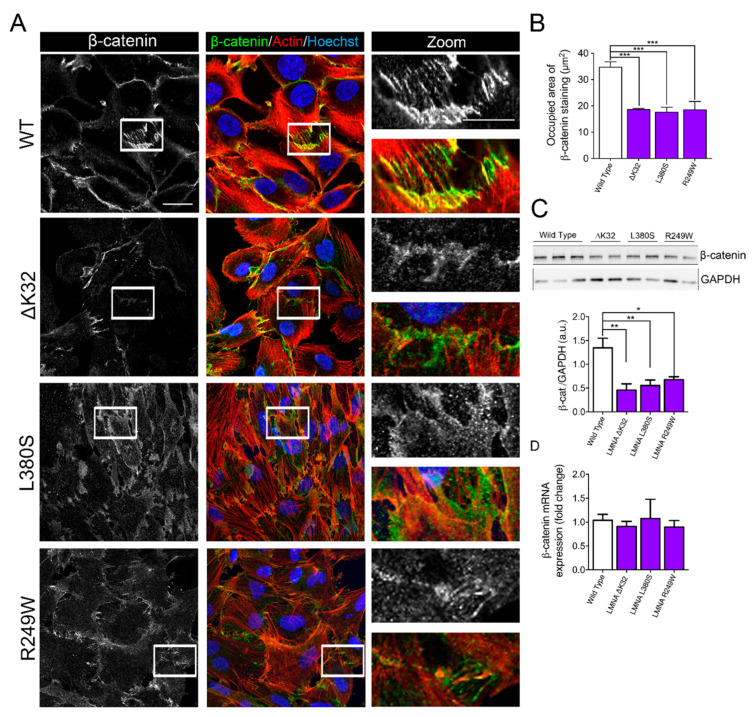
β-catenin in WT and mutant muscle cell precursors. (**A**). Confocal immunofluorescence images of F-actin (phalloidin, red) and β-catenin (white or green) in wild-type and *LMNA* mutant (ΔK32, L380S and R249W) mutant myogenic cell precursors. Nuclei are stained with Hoechst (blue). Scale bar: 20 µm. Zoomed region of cell-cell junctions are shown in left panels. Scale bar: 10 µm. (**B**). Quantification of the occupied area of β-catenin staining at cell-cell junctions. Pooled values of WT (WT1 and WT2) are presented. Values are means ± SEM from at least 4 different images/cell lines. *** *p* < 0.001 compared with WT. (**C**) Top: Representative Western blot of β-catenin and GAPDH in WT and mutant myoblasts. Bottom: Quantification of β-catenin protein levels expressed in arbitrary units (a.u.). GAPDH was used as a loading control. Pooled values of WT (WT1 and WT2) are presented. Values are means ± SEM, *n* ≥ 3 from at least 2 separate experiments. * *p* < 0.05, ** *p* < 0.01 compared with WT. (**D**) Relative mRNA expression of β-catenin (β-cat) normalized to RPLP0 and expressed as fold-changes. Pooled values of WT (WT1 and WT2) are presented. Values are means ± SEM, *n* = 3 separate experiments. There was no significant difference between cell lines.

**Figure 4 ijms-22-00306-f004:**
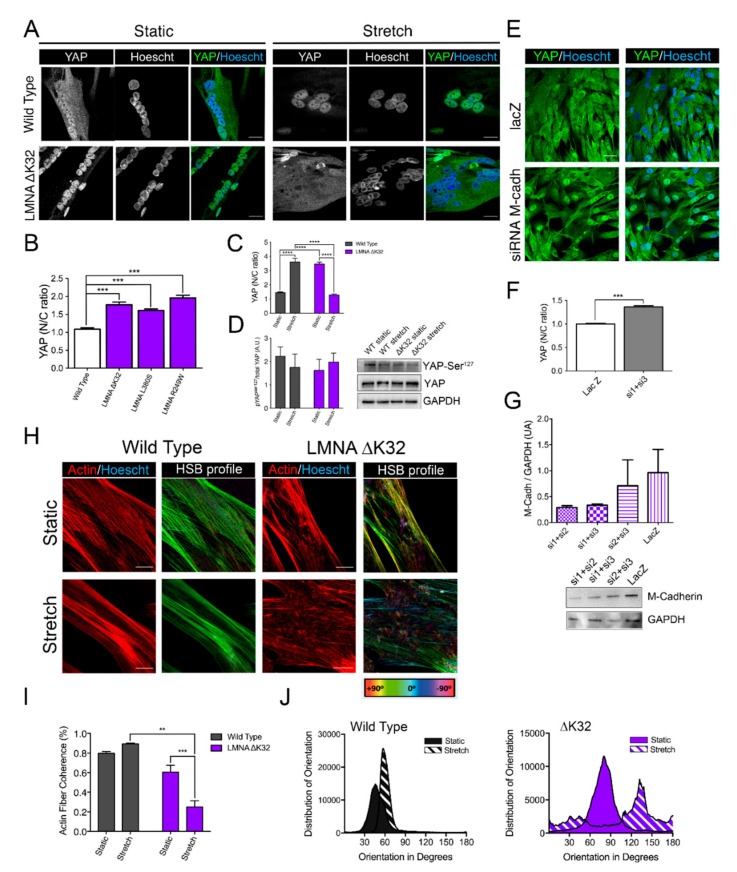
Adaptability of WT and *LMNA* ΔK32 myotubes to cyclic stretch (**A**) Yes-associated protein (YAP, in green) in WT and *LMNA* ΔK32 mutant myotubes (72 h differentiation) in static and after stretch. Nuclei are stained with Hoechst (blue). Scale bar: 20 µm (**B**) Quantification of YAP nucleo-cytoplasmic (N/C) ratio in WT and LMNA mutant myotubes. Pooled values of WT (WT1and WT2) are presented. Values are expressed as means ± SEM, *n* ≥ 60 cell in each cell line. *** *p* < 0.001 compared with WT. (**C**) Quantification of YAP nucleo-cytoplasmic (N/C) after cyclic stretch in myotubes. Pooled values of WT (WT1 and WT2) are presented. Values are expressed as means ± SEM, *n* ≥ 65 cell in each cell line. **** *p* < 0.0001 versus control values. (**D**) Left: Quantification of YAP Ser127 phosphorylation/total YAP protein. Values are expressed as means ± SEM. *n* = 3 separate experiments. Right: Representative Western blot of YAP Ser127, total YAP and GAPDH protein. (**E**) Confocal images of YAP (green) in WT treated with lacZ or siRNA against cadherin. Nuclei are stained with Hoechst (blue). Scale bar: 30 µm. (**F**) Quantification of YAP nucleo-cytoplasmic (N/C) ratio in WT treated with lacZ or siRNA against cadherin. Values are expressed as means ± SEM, *n* ≥ 180 cells in each group. *** *p* < 0.001 compared with lacZ-treated cells. (**G**) Top: Fold-change in M-cadherin protein levels. M-cadherin protein levels normalized to GAPDH. Values are means ± SEM, from 2 separate experiments. Bottom: Representative Western blot M-cadherin and GAPDH expression in confluent WT myoblasts treated with siRNA against M-cadherin or LacZ. (**H**) Actin (red) and HSB profile in WT and *LMNA* ΔK32 mutant myotubes in static and after stretch. Scale bar: 20 µm. (**I**) Actin fiber coherence in the dominant direction as determined by analysis of confocal images of myotubes stained fluorescently for actin (phalloidin) in ImageJ using OrientationJ plug-in. *n* = 3 images analyzed from 3 separate experiments per condition. ** *p* < 0.01, *** *p* < 0.001, compared with WT (**J**) Representative distribution of actin fiber orientation in static and stretched conditions for wild type and *LMNA* ΔK32 myotubes.

**Figure 5 ijms-22-00306-f005:**
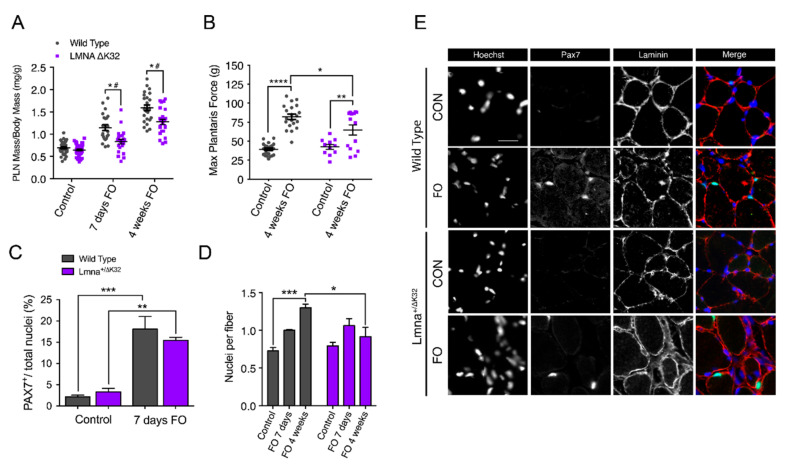
Functional and morphological abnormalities of Lmna+/ΔK32 mice to functional overload (FO). (**A**) Plantaris muscle mass normalized by body mass from WT and Lmna+/ΔK32 mice in control and after 7-days and 4-weeks FO. * *p* < 0.05 versus WT, # *p* < 0.05 versus control conditions. (**B**) Plantaris muscle maximal force from WT and Lmna+/ΔK32 mice in control and after 4 weeks FO. * *p* < 0.05, ** *p* < 0.005, **** *p* < 0.0001 (**C**) Quantification of Pax7+ cells as a percentage of total nuclei in control and after 7 days FO. Values are expressed as means ± SEM, ** *p* < 0.005, *** *p* < 0.001 versus control condition. (**D**) Quantification of nuclei per fiber from WT and Lmna+/ΔK32 mice in control and after 7 days and 4 weeks FO as determined by quantification of Hoechst-stained whole tissue sections by Myovision software. * *p* < 0.05 and *** *p* < 0.001. (**E**) Immunofluorescence images of PAX7+ (green) and laminin (red) in plantaris muscle section in WT and Lmna+/ΔK32 mice in control and after 7 days FO. Nuclei are stained with Hoechst (blue). Scale bar: 25 µm.

**Figure 6 ijms-22-00306-f006:**
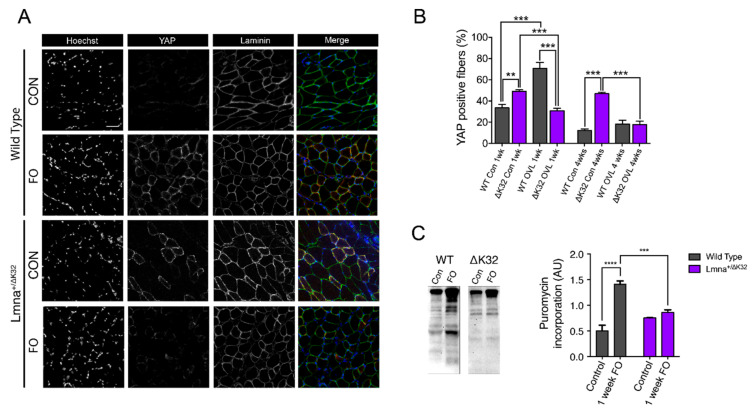
(**A**) Immunofluorescence images of YAP (green) and laminin (red) in control and 7-days FO plantaris muscles from WT and Lmna+/ΔK32 mice. Nuclei are stained with Hoechst (blue). Scale bar: 50 µm. (**B**) Quantification of yes-associated protein (YAP)+ fiber in control and 7 days FO plantaris muscles from WT and Lmna+/ΔK32 mice from at least 1000 fibers. ** *p* < 0.01 versus WT, *** *p* < 0.001 versus control conditions. (**C**) Representative Western blot and quantification of puromycin incorporation in control and 7 days FO plantaris muscles from WT and Lmna+/ΔK32 mice. *** *p* < 0.001, **** *p* < 0.0001 versus WT. Values are expressed as means ± SEM.

**Figure 7 ijms-22-00306-f007:**
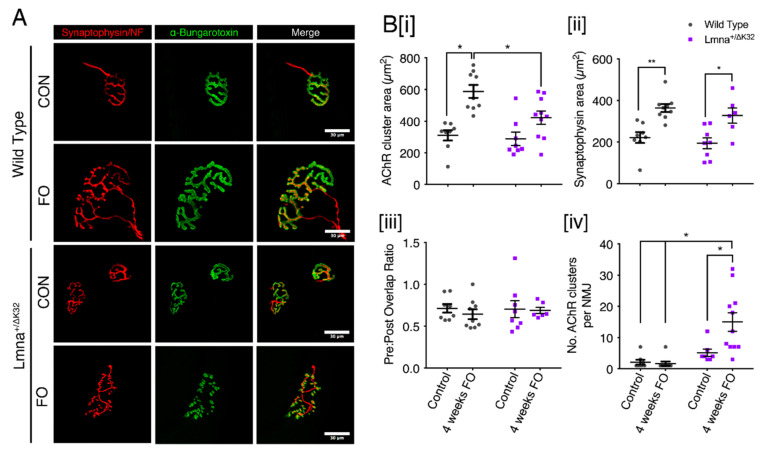
Neuromuscular junction defects of Lmna+/ΔK32 mice following FO. (**A**) Confocal immunofluorescence images of pre-synaptic structure (synaptophysin/neurofilament; red), post synaptic structure (a-bungarotoxin; green) and merged image. Scale bar: 30 µm. Values are expressed as means ± SEM (**B**) (**i**) Acetylcholine receptor cluster area in WT and Lmna+/ΔK32 mice in control conditions and following FO. * *p* < 0.05. (**ii**) synaptophysin area in WT and Lmna+/ΔK32 mice in control conditions and following FO. * *p* < 0.05, ** *p* < 0.005. (**iii**) number of acetylcholine receptor clusters per neuromuscular junction and (**iv**) Pre/post synapse overlap (i.e., synaptophysin/a-bungarotoxin) * *p* < 0.05.

**Figure 8 ijms-22-00306-f008:**
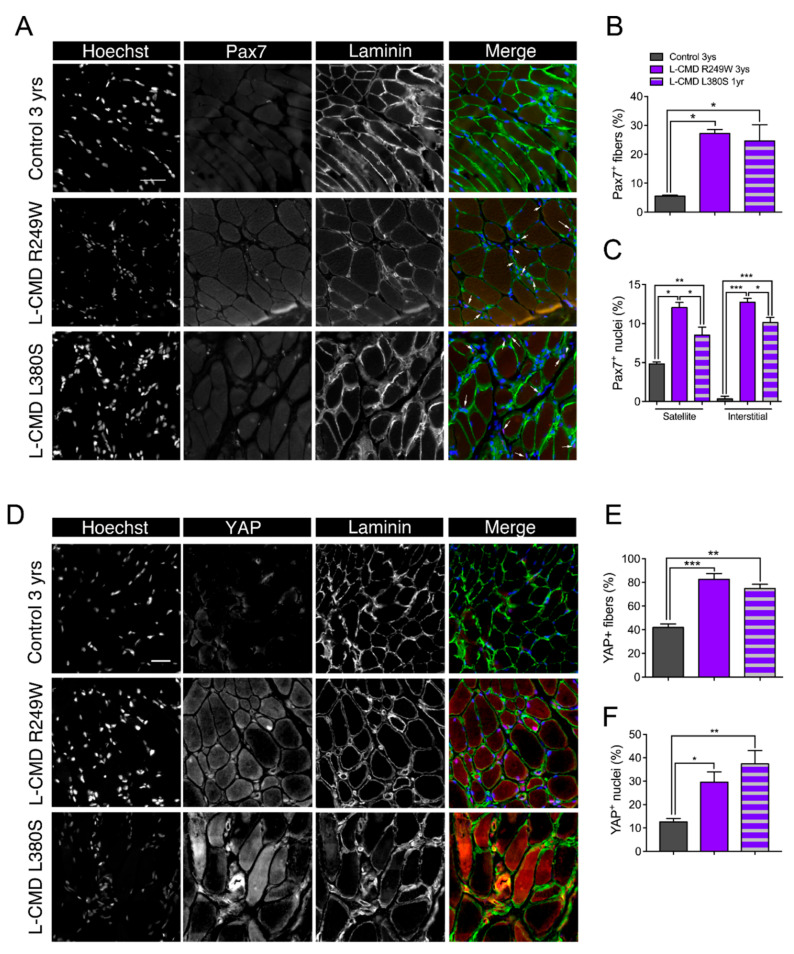
Histological data from muscle biopsies of patients with *LMNA*-CMD. (**A**) Immunofluorescence images of Pax7+ (green) and laminin (red) in muscle section from a control 3-year-old boy, a 3-year-old boy with R249W mutation and a 1-year-old boy with L380S mutation. Nuclei are stained with Hoechst (blue). Scale bar: 50 µm. (**B**,**C**) Quantification of Pax7+ cells per fiber and Pax7+ cells per nucleus in control and *LMNA*-CMD patients. Pax7+ cells in satellite or interstitial positions were determined. * *p* < 0.05, ** *p* < 0.01 and *** *p* < 0.001 *versus* control. (**D**) Immunofluorescence images of yes-associated protein (YAP, in green) and laminin (red) in muscle section from a control 3-year-old boy, a 3-year-old boy with R249W mutation and a 1-year-old boy with L380S mutation. Nuclei are stained with Hoechst (blue). Scale bar: 30 µm. (**E**,**F**) Quantification YAP+ cells per fiber and YAP+ cells per nucleus in control and *LMNA*-CMD patients. * *p* < 0.05, ** *p* < 0.01 and *** *p* < 0.001 *versus* control. Values are expressed as means ± SEM.

**Table 1 ijms-22-00306-t001:** Primer and siRNA sequences.

Human ribosomal protein lateral stalk subunit P0.	h-RPLPO	fw	CTCCAAGCAGATGCAGCAGA
		rev	ATAGCCTTGCGCATCATGGT
		rev	AAA-CCT-GAG-GCT-TCC-TCG-TC
siRNA1 against M-cadherin	siRNA1-cdh15	fw	CCC-UUG-AUG-ACA-UCA-AUG-A55
		rev	UCA-UUG-AUG-UCA-UCA-AGG-G55
siRNA2 against M-cadherin	siRNA2-cdh15	fw	CAU-CGC-CGA-CUU-CAU-CAA-U55
		rev	AUU-GAU-GAA-GUC-GGC-GAU-G55
siRNA3 against M-cadherin	siRNA3-cdh15	fw	GUG-AAC-CUC-AUC-UUU-GUA-U55
		rev	AUA-CAA-AGA-UGA-GGU-UCA-C55

**Table 2 ijms-22-00306-t002:** Characteristic of patients.

	Gender	*LMNA* Mutation	Muscle Biopsy/Age (Years)
P1	M	R249W	Deltoid/1 year
P2	F	L380S	Deltoid/3 years
P3	M	H222P	Biceps/59 years
C1	M		Unknown/4 days
C2	M		Unknown/3 years
C3	M		Quadriceps/33 years

## Data Availability

The data presented in this study are available on request from the corresponding author.

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
