# Peer review of "Lamin-Related Congenital Muscular Dystrophy Alters Mechanical Signaling and Skeletal Muscle Growth"

_ijms, 2020, doi:10.3390/ijms22010306_

Round 1

Reviewer 1 Report

Comment to authors

Owens et al., have investigated the mechanism underlying LMNA-CMD. Using a human muscle cell (MuSCs) differentiation model they report that mutant MuSCs fail to fuse to form myotubes and that this is related to degradation of M-cadherins leading to disruption of adherence junctions and M-cadherin/b-catenin/YAP signaling. As a consequence, LMNA-CMD mutant muscle fibers fail to organize actin fibers in response to mechanical stretch. In a LMNA-CMD mouse model they observe decreased ability to hypertrophy in response to functional overload and diminished myonuclear accretion. They also report decrease in YAP positive fibers in response to decreased protein synthesis in muscles of mutant mice. Finally, they study muscle biopsies from patients carrying LMNA-CMD mutations and report disorganized NMJs and mis-located satellite cells. They conclude that LMNA play an important role in mechano-transduction essential for muscle development and which is failing in LMNA-CMD. The new findings can potentially contribute to improved therapies in the future.

General comments:

The general impression is that the authors present vey interesting findings, which are important to move the field forward. It would be interesting to test the LMNA mutant cell lines using nuclear strain transfer experiments mentioned in ref 43!

However, the manuscript would gain from a thorough discussion about possible ways in which LMNA mutations cause perturbation of the M-cadherin/b-catenin/YAP signaling. Is the LINC complex involved? How can mutations in very different domains of laminA cause similar pathophysiology? It is also not clear why most studies were focused on the LMNADK32 mutation, when samples were only obtained from patients carrying the R249W and L380S mutations, respectively. It would also be interesting to have a discussion of why stretching the substrate have reciprocal effect on YAP distribution in wt versus mutant cells (Fig. 4A). Also, if YAP levels decrease with age (Supl. Fig.3) why are YAP intensities stronger in muscle fibers from patients compared to control (Fig. 8A)?  

A main concern is the way in which some of the data is presented. In many of the figure legends the number of individual experiments is not mentioned and in some cases the data is based on only 1 or 2 separate experiments. This is below standard scientific rigor and such results can at best only be regarded as preliminary. The authors must present data from at least 3 separate experiments in order to draw conclusions of the results or clearly motivate why this would not be necessary or impossible. Another concern is the risk of off-target effects in the RNAi experiments, which may lead to mis-interpretation (see specific comments).

Specific comments:

Introduction

Write about the LINC complex.

R88: …, these data our data….  Choose one or the other!

Results

R132: Here discuss why stretching the substrate results in reciprocal distribution in wt and mutant cells, respectively.

R238: …. lower to what observed… should be ….lower to what was observed…

Figures

Fig. 2C) Is statistics based on only one experiment?

Fig 2 B & D). Easier to read if controls are normalized to 1.

Fig.3 legend. Spelling mistake in next last row: ormalized should be normalized

Fig. 4.

  1. A) spell out Hoechst in left panel. The micrographs are not very clear and would gain from showing the individual channels side by side with the merged images. Are YAP levels higher in mutant cells compared to wt as was the case in muscle biopsies (Fig. 8)? This could be checked using WB.
  2. D) In the RNAi experiment 2 different siRNAs were combined, which makes the risk of off-target effects 2-fold higher. It is generally assumed that the off-target frequency of one siRNA is about one in thousand, which is high considering we have 20.000 genes. Thus, the risk is high that the observed phenotype is caused by unknown genes. Therefore, as a standard, two separate experiments have to be done using two independent siRNA. If the same phenotype is observed in both experiments, the off-target risk is one in a million, which is acceptable. Furthermore, was silencing effective? Should be demonstrated by IF or WB.

Fig. 5. State how many mice were used!

Fig. 6B) How as YAP+ cells scored and normalized from images in A? How many individual experiments?

Fig. 8. Why was not N/C ratio calculated as in Fig. 4? Why were not biopsies from LMNADK32 patients studied?

Suppl. Fig.3. Legend is mislabeled ! Where if E and F?

Materials and methods

4.1.6. Specify time for incubations.

4.2.2. Specify number of mice used

Overall this section should have same level of detail as 4.2.3, especially 4.1.7 lacks information.

4.2.5. Specify product number for antibodies used.

Author Response

Reviewer 1:

We thank this reviewer for their time and effort on reviewing our paper and sincerely appreciate the positive comments and insightful suggestions. The following are our point-by-point responses.

The manuscript would gain from a thorough discussion about possible ways in which LMNA mutations cause perturbation of the M-cadherin/b-catenin/YAP signaling. Is the LINC complex involved? How can mutations in very different domains of laminA cause similar pathophysiology?

Re: Thorough discussion about possible ways in which L-CMD mutations cause perturbations of the M-cadherin/β-catenin/YAP signaling have now been added. The involvement of the LINC complex and how different LMNA mutations could cause similar pathophysiology have been discussed (lines 81-88 and lines 298-305).

It is also not clear why most studies were focused on the LMNADK32 mutation, when samples were only obtained from patients carrying the R249W and L380S mutations, respectively.

Re: All experiments were performed on Lmna+/ΔK32 or 129S2/SvPasCrl LmnaH222P/H222P mice and corresponding WT mice. We focus on the mouse model carrying DK32 mutation because this is yet the only Lmna-CMD mouse model carrying human LMNA mutation available. Specifically, there are no available LmnaR249W and Lmna380S mouse models yet. This has been added (lines 175-176).

It would also be interesting to have a discussion of why stretching the substrate have reciprocal effect on YAP distribution in wt versus mutant cells (Fig. 4A).

Re: This has been done (lines 318-320).

Also, if YAP levels decrease with age (Supl. Fig.3) why are YAP intensities stronger in muscle fibers from patients compared to control (Fig. 8A)?  

Re: YAP levels decrease with age in controls. We reported altered YAP signaling in muscle biopsies from LMNA-CMD patients, which could explain the stronger YAP intensities in muscle fibers from patients compared to controls. These data supported our in vitro data.

A main concern is the way in which some of the data is presented. In many of the figure legends the number of individual experiments is not mentioned and in some cases the data is based on only 1 or 2 separate experiments. This is below standard scientific rigor and such results can at best only be regarded as preliminary. The authors must present data from at least 3 separate experiments in order to draw conclusions of the results or clearly motivate why this would not be necessary or impossible.

Re: The number of individual experiments has been added in each figure legends. We would like to reassure you that none of our data are based on only 1 experiment. We hope thatthis is now clearer in the text.

Another concern is the risk of off-target effects in the RNAi experiments, which may lead to mis-interpretation (see specific comments).

Re: Please see the answer in the specific comments below.

Specific comments:

Introduction

Write about the LINC complex.

This has been done (lines 63-69).

R88: …, these data our data….  Choose one or the other!

Re: This has been done (line 101)

Results

R132: Here discuss why stretching the substrate results in reciprocal distribution in wt and mutant cells, respectively.

Re: This has been done (lines 302-309).

R238: …. lower to what observed… should be ….lower to what was observed…

Re: This has been corrected (line 413)

Figures

Fig. 2C) Is statistics based on only one experiment?

Re: All experiments were based on at least n=3 from 2 separate experiments

Fig 2 B & D). Easier to read if controls are normalized to 1.

Re: This has been done.

Fig.3 legend. Spelling mistake in next last row: ormalized should be normalized

Re: This has been corrected

Fig. 4.

  1. A) spell out Hoechst in left panel. The micrographs are not very clear and would gain from showing the individual channels side by side with the merged images. Are YAP levels higher in mutant cells compared to wt as was the case in muscle biopsies (Fig. 8)? This could be checked using WB.

Re: The Figure 4 has been modified as recommended.

We agree that YAP quantification in control and LMNA patients would be interesting. However, nucleo‐cytoplasmic distribution of YAP/TAZ rather than the amount of YAP protein, is most probably a key determinant of their activity and is a major target of their regulation (for recent review see Wu et Guan, 2020 and Heng BC et al 2020). In addition, it is important to keep in mind that muscle biopsies from LMNA-CMD patients are very difficult to obtain and are very small being used mainly for the diagnosis of the patients. We do not have enough remaining tissue biopsies from these LMNA-CMD patients to perform protein extraction and to evaluate the amount of YAP protein in control and LMNA-CMD patients.

  1. D) In the RNAi experiment 2 different siRNAs were combined, which makes the risk of off-target effects 2-fold higher. It is generally assumed that the off-target frequency of one siRNA is about one in thousand, which is high considering we have 20.000 genes. Thus, the risk is high that the observed phenotype is caused by unknown genes. Therefore, as a standard, two separate experiments have to be done using two independent siRNA. If the same phenotype is observed in both experiments, the off-target risk is one in a million, which is acceptable. Furthermore, was silencing effective? Should be demonstrated by IF or WB.

Re: We tested different combinations of siRNA against M-cadherin in 2 separate experiments and analyzed the protein expression versus lacZ and MOCK experiments. Combination of siRNA1+ siRNA 2 and siRNA1+siRNA3 induced a nearly 70% reduction in protein expression compared to the LacZ value, whereas the combination of siRNA2+siRNA3 was ineffective. Data have been added in Revised Figure 4.

Fig. 5. State how many mice were used!

Re: The number of mice has been added (para 4.2).

Fig. 6B) How as YAP+ cells scored and normalized from images in A? How many individual experiments?

Re: Methodological information has been added, as recommended (lines 527-532).

Fig. 8. Why was not N/C ratio calculated as in Fig. 4? Why were not biopsies from LMNADK32 patients studied?

Re: There were no biopsies from LMNA-DK32 patients because we do not have any muscle biopsies from these patients. This is a very rare disease whose diagnosis is not based on muscle biopsy. Thus, muscle biopsies from LMNA-CMD patients are extremely rare and very difficult to obtain.

Suppl. Fig.3. Legend is mislabeled ! Where if E and F?

Re: This has been corrected.

Materials and methods

4.1.6. Specify time for incubations.

Re: This has been done.

4.2.2. Specify number of mice used

Re: Overall this section should have same level of detail as 4.2.3, especially 4.1.7 lacks information.

Number of mice used has been added (lines 450-463) and additional information have been added in 4.2.3, as recommended.

4.2.5. Specify product number for antibodies used.

Re: This has been done.

Reviewer 2 Report

In this manuscript Owens and colleagues investigate the impact of LMNA mutations causing congenital muscular dystrophy (CMD) on muscle stem cells (MuSCs) and their ability to form myotubes. They show that CMD mutations in MuSCs impair myotube fusion and that these have compromised ability to respond to cyclic stretch stimuli. Importantly, authors supplement these in vitro findings with in situ findings of abrogated hypertrophy response in plantaris muscles coupled with changes in NMJs of mutated mice and show consistent data in patients. Thus, this increases the physiological relevance of authors findings, the novelty and interest to readers.

The data in general support author’s conclusion, however, there are some missing links on the mechanistic level that need to be clarified. Authors claim that impaired cell-cell junctions in LMNA mutated MuSCs cause defective myotube fusion leading to compromised actin mechanoresponse and YAP signaling upon exposure to cyclic stretch. Authors need to distinguish between their findings in MuSCs and differentiated myotubes. Is there a change in YAP localization and signaling in mutated MuSC and what is the effect of M-cadherin downregulation in Wt MuSCs? On the other hand, how does the compromised mechanoresponse and YAP mislocalization in myotubes from mutated LMNA MuSCs correlate to changes in cell-cell junctions under cyclic stretch. Authors have previously published increased F-actin/G-actin ratio in mutated LMNA myoblasts (Bertrand et al.,2014). Can myotube fusion defect and upregulated YAP signaling in LMNA mutated MuSCs be ameliorated by using inhibitors of actin polymerization such as ROCK-inhibitor? Or by M-cadherin overexpression? Answering these questions would provide a clearer picture why there is an upregulated YAP signaling under control conditions in mutated LMNA cells that appears unresponsive to further activation upon functional overload.

Specific comments

  1. The activation status of YAP using e.g. Ser 112 YAP antibodies but also total protein levels of this protein under control conditions and after stretch should be tested using immunoblotting from cell extracts (Figures 4 and Figure 6).
  2. Is there a change in Pax7+ cells in untreated conditions in other two LMNA mutations LMNA p.Arg249Trp (LMNAR249W) and LMNA p.Leu380Ser (LMNAL380S) (Figure 5C)? This is important in order to correlate these data to patients in Figure 8.
  3. Figure 4C and D, control showing M-cadherin knockdown efficiency is missing.
  4. Figure 6A. Authors should provide high-power images showing YAP localization at the periphery of muscle cells.

Minor comments

  1. Authors should provide information on how the MuSCs were generated and details on their characterization.
  2. Statistical evaluation is not sufficiently explained. It is not clear whether authors used non-parametric tests as this should be the case in figures such as fold values and percentages. Or otherwise authors should provide details in Material and methods  how these were evaluated.
  3. The introduction chapter needs further explanation on YAP/TAZ signaling and how it is regulated by stretch or actin cytoskeleton. Regulation by actin cytoskeleton is explained in Results section quite late (lane 150) but the citation for this statement is not included. Furthermore, authors previous data on YAP signaling in muscle precursor should be explained in more detail in introduction.

Author Response

Thank you for your kind review and suggestions. 

Please check attached point-to-point reply document. 

Round 2

Reviewer 2 Report

The authors have addressed the raised concerns, provided controls, and performed some but not all additional experiments to support the authors major conclusions.

Small issues that need to be resolved

Edited figures appear twice such as Figure 6 and Figure 7. In Figure 6A indicate the scale bars for different panels. Missing scale bars in other Figures should be added (eg. Figure 5). Is the magnification for Hoechst staining and laminin/YAP the same in Figure 6A? If not explain in Figure legend and add different scale bars.